# Precision Medicine for the Management of Therapy Refractory Colorectal Cancer

**DOI:** 10.3390/jpm10040272

**Published:** 2020-12-11

**Authors:** Hossein Taghizadeh, Robert M. Mader, Leonhard Müllauer, Friedrich Erhart, Alexandra Kautzky-Willer, Gerald W. Prager

**Affiliations:** 1Department of Medicine I, Clinical Division of Oncology, Medical University of Vienna, 1090 Vienna, Austria; seyed.taghizadehwaghefi@meduniwien.ac.at (H.T.); robert.mader@meduniwien.ac.at (R.M.M.); 2Comprehensive Cancer Center Vienna, 1090 Vienna, Austria; leonhard.muellauer@meduniwien.ac.at; 3Clinical Institute of Pathology, Medical University Vienna, 1090 Vienna, Austria; 4Department of Internal Medicine, Amstetten Region State Clinic, 3300 Amstetten, Austria; Friedrich.Erhart@amstetten.lknoe.at; 5Department of Medicine III, Gender Medicine Unit, Medical University of Vienna, 1090 Vienna, Austria; alexandra.kautzky-willer@meduniwien.ac.at

**Keywords:** molecular oncology, precision medicine, colorectal cancer, targeted therapy, molecular profiling

## Abstract

In this analysis, we examined the efficacy, feasibility, and limitations of molecular-based targeted therapies in heavily pretreated metastatic colorectal cancer (mCRC) patients after failure of all standard treatments. In this single-center, real-world retrospective analysis of our platform for precision medicine, we mapped the molecular profiles of 60 mCRC patients. Tumor samples of the patients were analyzed using next-generation sequencing panels of mutation hotspots, microsatellite instability testing, and immunohistochemistry. All profiles were reviewed by a multidisciplinary team to provide a targeted treatment recommendation after consensus discussion. In total, we detected 166 mutations in 53 patients. The five most frequently found mutations were *TP53, KRAS, APC, PIK3CA,* and *PTEN*. In 28 cases (47% of all patients), a molecularly targeted therapy could be recommended. Eventually, 12 patients (20%) received the recommended therapy. Six patients (10%) had a clinical benefit. The median time to treatment failure was 3.1 months. Our study demonstrates the feasibility and applicability of using targeted therapies in daily clinical practice for heavily pretreated mCRC patients. This could be used as a targeted treatment option in half of the patients.

## 1. Introduction

Colorectal cancer (CRC) is one of the most frequent cancer types and is a major cause of mortality and morbidity. According to GLOBOCAN 2018, CRC is the fourth most common cancer disease throughout the world, equally affecting both men and women, with over 2 million new cases in 2018 [1]. It accounts for approximately 1 million deaths annually and 11% of all cancer deaths, ranking as the third most common cause of cancer death [1]. CRC particularly affects developed countries where inhabitants follow a western lifestyle that bears important risk factors for the carcinogenesis of CRC including alcohol intake, tobacco use, immoderate red and processed meat consumption, low intake of fiber, obesity, and a sedentary lifestyle [2].

In recent years, considerable effort has been made to explore the complex tumor biology of CRC and to expand and enrich the therapeutic armamentarium with new therapeutic agents. The therapeutic landscape for the management of metastatic CRC (mCRC) is rapidly evolving.

The development and application of monoclonal antibodies and tyrosine kinase inhibitors in addition to systemic cytotoxic (poly)chemotherapy have significantly improved the prognosis, median overall survival, and quality of life of mCRC patients [3,4].

Despite diagnostic and therapeutic advances in the management of mCRC, the 5-year survival rate for mCRC is approximately 14%, with a median overall survival of 30 months [5,6]. Moreover, after failure of standard therapy lines, therapeutic options are limited.

One way to offer treatment concepts for therapy refractory mCRC would be to analyze the molecular profile of tumors to identify actionable pathologic molecular alterations to develop an individually coordinated therapy plan. This individually tailored, tissue-agnostic molecular-based treatment approach is referred to as precision medicine in oncology or simply precision oncology [7,8].

In the last few years, more and more targeted therapy agents have been introduced for the management of several cancer diseases, such as trastuzumab in human epidermal growth factor receptor 2 (HER2 positive) breast cancer or gastric cancer [9,10], imatinib in in KIT+ gastrointestinal stromal tumor (GIST) [11], and B-rapidly accelerated fibrosarcoma (BRAF)-directed therapy with vemurafenib or dabrafenib/trametinib in melanoma [12].

Thus, exploring the molecular profile of mCRC may aid in the development of molecular targeted therapies and allow their efficacy to be tested.

In this study, we conducted a retrospective subgroup analysis of all 60 patients with advanced therapy refractory mCRC that had been enrolled and profiled via our special platform for precision medicine of the Comprehensive Cancer Centre of the Medical University of Vienna (CCC-MUV).

We sought to map the molecular profiles of mCRC to identify and target specific molecular alterations. We discuss the challenges, limitations, and the time to treatment failure (TTF) of precision medicine approaches in this patient group.

## 2. Methods

### 2.1. Patients and Design of the Precision Medicine Platform

All patients with heavily pretreated advanced metastatic CRC who had progressed to all standard treatment options and had undergone molecular profiling from June 2013 to June 2020 were included in this retrospective single-center study. This study was conducted at the Clinical Division of Oncology of the tertiary care university hospital Medical University of Vienna. Cancer patients refractory to all standard therapies were eligible for inclusion in our precision medicine platform, provided that tissue samples for molecular profiling were available. The specimens were either obtained by fresh tumor biopsy performed by physicians at the Department of Interventional Radiology or were provided by the archives of the Department of Pathology when tumor biopsy was not feasible. Patients had to have an Eastern Cooperative Oncology Group (ECOG) performance status of ≤1. All patients in this analysis had to be at least 18 years at the time of molecular analysis and had to provide informed consent before inclusion in our platform. Our precision medicine platform is not a clinical trial but intends to provide targeted therapy recommendations to patients where no standard antitumoral treatment is available. This analysis was approved by the Institutional Ethics Committee of the Medical University of Vienna (Nr. 1039/2017). The General Hospital of Vienna directly covered all costs for molecular profiling and targeted therapy provided the cancer patients had no further standard treatment options.

### 2.2. Evaluation of Outcome and Follow-Up

All patients with heavily pretreated advanced metastatic CRC who had progressed to all standard treatment options were confirmed by the response evaluation criteria in solid tumors 1.1 (RECIST 1.1) criteria [13]. These international criteria provide a basis for standardized and objective assessment of the change in tumor burden during treatment. The criteria distinguish four types of change:Complete response (CR): All target lesions disappearPartial response (PR): The sum of the longest diameter of target lesions decrease at least by 30%Stable disease (SD): Neither sufficient shrinkage to qualify for PR nor sufficient increase to qualify for PDProgressive disease (PD): The sum of the longest diameter of target lesions increase at least by 20%. PD means the tumor has become resistant to the therapy and, thus, the therapy has failed.

Follow-up was done every 8 to 12 weeks for outcome evaluation by radiological assessment depending on the respective therapy. If the patient did not appear on the follow-up date, we searched our electronic data processing system that is linked to the national death register to check and ascertain the death of the patient in the meantime.

### 2.3. Tissue Samples

Formalin-fixed, paraffin-embedded tissue samples from patients with metastatic CRC who had progressed through all standard therapy regimens were obtained from the archives of the Department of Pathology, Medical University of Vienna, Austria.

### 2.4. Cancer Gene Panel Sequencing

DNA was extracted from paraffin-embedded tissue blocks with a QIAamp Tissue KitTM (Qiagen, Hilden, Germany). From each tissue sample, 10 ng of DNA was provided for sequencing. The DNA library was created by multiplex polymerase chain reaction with the Ion AmpliSeq Cancer Hotspot Panel v2 (Thermo Fisher Scientific, Waltham, MA, USA), which covers mutation hotspots of 50 genes. The panel includes driver mutations, oncogenes, and tumor suppressor genes. In mid-2018, the gene panel was expanded using the 161-gene next-generation sequencing panel of Oncomine Comprehensive Assay v3 (Thermo Fisher Scientific, Waltham, MA, USA), which covers genetic alterations, gene amplifications, and gene fusions. The Ampliseq cancer hotspot panel was sequenced with an Ion PGM (Thermo Fisher) and the Oncomine Comprehensive Assay v3 on an Ion S5 sequencer (Thermo Fisher Scientific, Waltham, MA, USA). The generated sequencing data were analyzed afterwards with the help of Ion Reporter Software (Thermo Scientific Fisher). We referred to the BRCA Exchange, ClinVar, COSMIC, dbSNP, OMIM, and 1000 genomes for variant calling and classification. The variants were classified according to a five-tier system comprising pathogenic, likely pathogenic, uncertain significance, likely benign, or benign modifiers. This classification was based on the standards and guidelines for the interpretation of sequence variants of the American College of Medical Genetics and Genomics [14]. The pathogenic and likely pathogenic variants were taken into consideration for the recommendation of targeted therapy.

### 2.5. Immunohistochemistry

Immunohistochemistry (IHC) was performed using 2-μm-thin tissue sections that were read by a Ventana Benchmark Ultra stainer (Ventana Medical Systems, Tucson, AZ, USA). The following antibodies were applied: anaplastic lymphoma kinase (ALK) (clone 1A4; Zytomed, Berlin, Germany); CD20 (clone L26; Dako); CD30 (clone BerH2; Agilent Technologies, Vienna, Austria); DNA mismatch repair (MMR) proteins including MLH1 (clone M1, Ventana Medical Systems), PMS2 (clone EPR3947, Cell Marque, Rocklin, CA, USA), MSH2 (clone G219-1129, Cell Marque), and MSH6 (clone 44, Cell Marque); epidermal growth factor receptor (EGFR) (clone 3C6; Ventana); estrogen receptor (clone SP1; Ventana Medical Systems); human epidermal growth factor receptor 2 (HER2) (clone 4B5; Ventana Medical Systems); HER3 (clone SP71; Abcam, Cambridge, UK); C-kit receptor (KIT) (clone 9.7; Ventana Medical Systems); MET (clone SP44; Ventana); NTRK (clone EPR17341, Abcam); phosphorylated mammalian target of rapamycin (p-mTOR) (clone 49F9; Cell Signaling Technology, Danvers, MA, USA); platelet-derived growth factor alpha (PDGFRA) (rabbit polyclonal; Thermo Fisher Scientific); PDGFRB (clone 28E1, Cell Signaling Technology); programmed death-ligand 1 (PD-L1) (clone E1L3N; Cell Signaling Technology till mid-2018, as of mid-2018 the clone BSR90 from Nordic Biosite, Stockholm, Sweden is used); progesterone receptor (clone 1E2; Ventana); phosphatase and tensin homolog (PTEN) (clone Y184; Abcam); and ROS1 (clone D4D6; Cell Signaling Technology).

To assess the immunostaining intensity for the antigens EGFR, p-mTOR, PDGFRA, PDGFRB, and PTEN, a combinative semiquantitative score for immunohistochemistry was used. The immunostaining intensity was graded from 0 to 3 (0 = negative, 1 = weak, 2 = moderate, and 3 = strong). To calculate the score, the intensity grade was multiplied by the percentage of corresponding positive cells: (maximum 300) = (% negative × 0) + (% weak × 1) + (% moderate × 2) + (% strong × 3).

The immunohistochemical staining intensity for HER2 was scored from 0 to 3+ (0 = negative, 1+ = negative, 2+ = positive, and 3+ = positive) pursuant to the scoring guidelines of the Dako HercepTestR from the company Agilent Technologies (Agilent Technologies, Vienna, Austria). In case of HER2 2+, a further test with HER2 in situ hybridization was performed to verify HER2 gene amplification.

Estrogen receptor and progesterone receptor staining were graded according to the Allred scoring system from 0 to 8. MET staining was scored from 0 to 3 (0 = negative, 1 = weak, 2 = moderate, and 3 = strong) based on a paper by Koeppen and coworkers. [15].

For PD-L1 protein expression, the tumor proportion score, which is the percentage of viable malignant cells showing membrane staining, was calculated. In addition, since 2019, the expression has also been determined by a combined positive score.

ALK, CD30, CD20, and ROS1 staining were classified as positive or negative based on the percentage of reactive tumor cells, however without graduation of the staining intensity. In ALK- or ROS1-positive cases, the presence of possible gene translocation was evaluated by fluorescence in situ hybridization (FISH).

All antibodies used in this study were validated and approved at the clinical institute of pathology of the Medical University of Vienna and are used in routine IHC staining for clinical purposes. The antibodies have been validated by proper positive and negative tissue controls and by non-IHC methods, such as immunoblotting and flow cytometry, to detect the respective epitopes of the antigens. For control, the use of the antibodies was optimized in terms of intensity, concentration, signal/noise ratio, incubation times, and blocking. The negative control involved omitting the primary antibody and substituting an isotype-specific antibody and serum at the exact same dilution and laboratory conditions as the primary antibody to preclude unspecific binding.

For positive control, the antibodies were shown not to cross-react with closely related molecules of the target epitope.

The status of microsatellite instability-high (MSI-H) was analyzed by the MSI Analysis System, version 1.1 (Promega Corporation, Madison, WI, USA).

### 2.6. Fluorescence in situ Hybridization (FISH)

FISH was only applied in selected cases to verify PTEN loss. FISH was performed with 4-μm-thick formalin-fixed, paraffin-embedded tissue sections. The following FISH probes were utilized: PTEN (10q23.31)/centromere 10 (ZytoVision, Bremerhaven, Germany). Two hundred cell nuclei were evaluated per tumor. The PTEN FISH was considered positive for PTEN gene loss with ≥30% of cells with only one or no PTEN signal. A chromosome 10 centromere FISH probe served as a control for the ploidy of chromosome 10.

### 2.7. Multidisciplinary Team for Precision Medicine

After thorough examination of the molecular profile of each tumor sample by a qualified molecular pathologist, the results were reviewed by a multidisciplinary team (MDT) that met every other week.

Members of the MDT included molecular pathologists, radiologists, clinical oncologists, surgical oncologists, and basic scientists. The MDT recommended targeted therapy based on the specific molecular profile of each patient, which included established pathological parameters. The targeted therapies included tyrosine kinase inhibitors, checkpoint inhibitors (e.g., anti-PD-L1 monoclonal antibodies), and growth factor receptor antibodies with or without endocrine therapy. The treatment recommendations by the MDT were prioritized using the level of evidence from high to low according to phase III to phase I trials.

In cases where more than one druggable molecular aberration was identified, the MDT recommended a therapy regimen to target as many molecular aberrations as possible, with special consideration given to the toxicity profile of each antitumoral agent and its potential interactions. Since all patients were given all available standard treatment options for their cancer disease prior to inclusion in our precision medicine platform, nearly all targeted agents suggested had off-label use. If the tumor profile and clinical characteristics of a patient met the requirements to be enrolled in a recruiting clinical trial for targeted therapies at our cancer center, patients were asked whether they wanted to participate in the respective trial, and trials adhered to ethical and regulatory guidelines.

### 2.8. Study Design and Statistics

The Fisher’s exact test was employed to explore potential gender-specific differences regarding the therapy recommendation rate and the molecular profile. Student’s *t*-test was applied to test differences in the outcome between tumor samples that were obtained during tumor biopsy versus specimens that were obtained during surgical resection. A *p*-value of less than 0.05 was considered to be statistically significant. For statistical analysis, the software package IBM SPSS Statistics Version 26 was used.

## 3. Results

### 3.1. Patient Characteristics

From the initiation of our platform for precision medicine in June 2013 until June 2020, we identified 60 patients with therapy refractory mCRC with no further standard treatment option available. These patients were all included in this subgroup analysis of our platform. All 60 patients were Caucasian, including 42 men (70%) and 18 women (30%). The cohort of mCRC patients comprised 47 patients (78%) with left-sided CRC and 13 patients (22%) with right-sided CRC (see Figure 1 for the patients’ flow and Table 1 for the patient characteristics).

The median age at first diagnosis was 54.9 years (range: 16.9 to 81.2 years), and the median age at the time of molecular profiling was 57.9 years (range: 19.3 to 84.3 years). Tumor tissue used for molecular profiling was obtained by biopsy in 25 patients (42%) or during surgical treatment in the other 35 patients (58%). Biopsy was performed after failure of all standard treatment options. The median time interval between resection and molecular analysis of the tumor tissue was 11.9 months (range: 1–46 months). The median time interval between biopsy and completion of the molecular analysis of the tumor tissue was 31 days (range: 15–56 days). The median turnaround time between the initiation of molecular profiling and discussion by the MDT and molecular-based therapy initiation for the 12 patients who received the targeted therapy was 30 and 42 days, respectively. The median time interval between the initiation of molecular profiling and discussion of the MDT and molecular-based therapy initiation for the other patients (*n* = 48) was 29 days.

Twenty-one patients experienced a disease relapse. All of the patients had metastases, mainly in the liver, lungs, and bones. Seventeen patients had additional intraperitoneal dissemination of the CRC, causing peritoneal carcinomatosis. The patients received a median of three lines of prior palliative systemic chemotherapy, ranging from two to six lines. The palliative therapy regimens included FOLFOX + cetuximab, FOLFOX + bevacizumab, FOLFIRI + cetuximab, FOLFOX + panitumumab, FOLFIRI + panitumumab, FOLFOXIRI + bevacizumab, FOLFIRI + bevacizumab, FOLFIRI + aflibercept, FOLFIRI + ramucirumab, regorafenib, trifluridine/ tipiracil, and raltitrexed + oxaliplatin. Twenty-six patients (43%) had received at least four lines of palliative chemotherapy prior to molecular profiling.

### 3.2. Molecular Profile

In total, we detected 166 mutations in 53 patients (88%). The five most frequent mutations were *TP53* (*n* = 36; 60.0%), *KRAS* (*n* = 29; 48.3%), *APC* (*n* = 15; 25.0%), *PIK3CA* (*n* = 9; 15%), and *PTEN* (*n* = 8; 13.3%), which accounted for more than half of all mutations (58.4%). No mutations were detected in seven (12%) patients with our sequencing panel (Table 2). Seven gene fusions were identified in five patients: *FGFR3-TACC3* (*n* = 2), *FNDC3B-PIK3CA*, *SND1-BRAF*, *EIF3E-RSPO2*, *PTPRK-RSPO3*, and *WHSC1L1-FGFR1*. Moreover, we detected eight gene amplifications in six different tumor specimens, including *CCND2 (n = 3), FLT3 (n = 3), FGFR1,* and *MYC.*

Further, IHC revealed common expressions of phosphorylated mTOR and EGFR in 49 (82%) and 45 (75%) patients, respectively. The median IHC scores of mTOR and EGFR were 100 and 90, respectively. Ten patients (17%) had high levels of phosphorylated mTOR expression with an mTOR score between 200 and 300. EGFR expression was between 200 and 300 in nine patients (15%). In our cohort, two patients (3%) were HER2-positive and seven patients (12%) were HER3-positive. IHC identified six patients (10%) with a loss of PTEN, which was subsequently verified and characterized by FISH as heterozygous PTEN deletions. High expressions levels were also observed for MET (*n* = 28) and PDGFRA (*n* = 15). Four patients (7%) were given a status of MSI-H. In four patients, the PD-L1 combined positive score was ≥1. Three patients displayed a weak KIT expression. The expression of other markers was not observed. IHC and FISH could not be performed for one male patient due to insufficient tumor material. See Figure 2a–n.

### 3.3. Therapy Recommendations and Outcome

In 28 cases (47% of all patients), a molecularly targeted therapy was recommended. The other 32 patients (53%) did not qualify for targeted therapy due to the lack of actionable molecular targets. In over two-thirds of all recommendations (*n* = 20/28, 71.0%), the molecular-driven treatment approach was mainly derived from the molecular characteristics determined by immunohistochemistry. The 28 recommended targeted treatments included everolimus, pembrolizumab, nintedanib, cetuximab, vismodegib, vemurafenib, afatinib and trastuzumab combined with lapatinib, crizotinib, erlotinib, and sunitinib plus capecitabine. Table 3 describes the rationale for the recommended targeted therapy approaches. Eventually, 12 patients (20%) received the recommended targeted therapy. One patient with the KIT mutation was included in the clinical phase II trial SUNCAP and was treated with sunitinib and capecitabine. Another patient died prior to restaging. Finally, ten patients underwent radiological assessment (see Table 4). Four patients (7%) experienced progressive disease. Four patients with MSI-H status were given pembrolizumab and achieved a partial response (*n* = 2), complete response, and stable disease, respectively. Two patients treated with other targeted agents achieved a stable disease. Thus, the disease control rate was 10%. The three patients who achieved therapy response and one of the three patients who had stable disease experienced also an improvement in their quality of life due to an improvement in tumor pain intensity. The molecularly targeted therapies applied were pembrolizumab (*n* = 4), trastuzumab (*n* = 2), everolimus plus bevacizumab (*n* = 2), afatinib, everolimus, sunitinib plus capecitabine, and everolimus plus raltitrexed (see Table 3 and Table 4 for further information). The median time to treatment failure (TTF) in patients who received the targeted therapy was 3.1 months (range: 0.3–30.6 months; see Figure 3 and Table 4). The median overall survival (mOS) of these 12 patients after initial diagnosis of mCRC was 50.1 months. The mOS after initiation of targeted therapy was 10.9 months (see Figure 4a,b). Three patients were lost to follow-up after the suggestion of molecular-driven targeted therapy. Thirteen patients (22%) did not receive the offered targeted therapy. Reasons for not applying the recommended targeted agent included the following: rapid deterioration of performance status (*n* = 10), death of patient (*n* = 1), the treating oncologist favoring another treatment regimen due to the clinical overall situation of the patients, or patients’ refusal of any further treatment including targeted therapy options (*n* = 2).

Three tumor specimens from the ten patients who underwent radiological assessment were obtained during a conventional tumor biopsy. Two of these three patients experienced progressive disease, and one patient achieved stable disease. The tumor sample of the remaining seven patients was yielded during surgical resection of the primary tumor. We found no significant difference between tumor samples obtained during tumor biopsy versus surgical resection in terms of TTF (*p* = 0.319) and mOS (*p* = 0.396).

According to the Fisher’s exact test, the gender-specific differences regarding the 28 targeted therapy recommendations were not statistically significant (*p* = 0.516).

## 4. Discussions

In our study, we recommended 28 molecularly targeted therapies based on the respective individual molecular profile of heavily pretreated mCRC patients. Thus, precision medicine approaches were found to be feasible and implementable in daily clinical routine in approximately half of the patients who had no further standard treatment option. In over two-thirds of all recommendations, the molecular-based targeted treatment approach was mainly derived from the molecular characteristics determined by immunohistochemistry. This fact underlines the major clinical relevance of immunohistochemistry in precision medicine as immunohistochemistry and next-generation sequencing complement each other. This could be very useful as sequencing panels are updated and enlarged routinely, adding valuable information for treatment decision making.

However, one important limitation of this study was that other parts of the molecular portrait were not analyzed. The molecular profile of a tumor is intricate and complex and goes beyond these two techniques. Comprehensive mapping of the molecular profile is multilayered and multi-faceted and includes many other aspects, including genomics, epigenomics, transcriptomics, proteomics including phospho- and glycoproteomics, metabolomics, epigenetics, and microbiomics [16]. The processing and integration of these extremely large quantities of data and their translation into targeted therapy recommendations is a grand challenge that scientists and clinicians are confronted with. There are close links among all involved disciplines to achieve common objectives. Further, CRC is characterized by highly dynamic and complex molecular intratumoral and intertumoral heterogeneity that changes both temporally and spatially [16,17,18,19,20,21]. The tumor tissue used for molecular profiling was obtained by biopsy in 25 patients (42%) or during surgical treatment in the other 35 patients (58%). Biopsy was performed after failure of all standard treatment options. Whenever possible, we used metastatic tissue for molecular profiling, which was particularly suited when fresh biopsies were obtained. If this approach was not feasible, e.g., if the anatomic site was not suitable for biopsy, we used the information obtained from the primary tumor site. Despite potential spatial heterogeneity, we assumed that most of the genetic aberrations in the primary cancer were also present at the metastatic site. Studies have shown that there is a high biomarker concordance between primary colorectal cancer and its metastases [22]. We found no significant difference between tumor samples obtained during tumor biopsy versus surgical resection in terms of TTF. However, the number of patients (n = 10) who underwent radiological response assessment after application of the targeted therapy was too limited to examine the influence of biomarker concordance on the outcome. Thus, further clinical trials and studies are required to examine the degree of biomarker concordance between primary cancer and metastases.

In our cohort, the five most frequent mutations, *TP53, KRAS, APC, PIK3CA,* and *PTEN*, together accounted for more than 50% percent of all detected mutations. Except for PIK3CA mutations, there are still no molecularly targeted therapies that directly target the mutations in *TP53*, *KRAS*, *APC*, and *PTEN*. Thus, there is an unmet clinical need for the inhibition of these genetic aberrations. The rest of the detected mutations were of low frequency (below 10%) and reflect the well-known molecular heterogeneity and diversity of CRC. The detected genetic aberrations are in line with the results of previous studies [23,24,25].

Moreover, a growing body of evidence shows that the antitumoral therapy itself may affect, influence, and drive tumor molecular evolution [26,27,28]. A prime example of this phenomenon is the recommendation of cetuximab for two patients who were initially KRAS-mutated and were therefore not treated with an anti-EGFR therapy; however, at the time of molecular profiling, they had developed the RAS wildtype [29].

One way to monitor the dynamic molecular landscape of cancer disease would be the utilization of real-time liquid biopsy to adapt antitumoral therapy according to the current molecular portrait [30]. To this end, the multicenter clinical phase II trial MoliMor (EudraCT number: 2019–003714–14) is evaluating the efficacy and safety of intermittent addition of cetuximab to a FOLFIRI-based first-line therapy to patients with RAS-mutant mCRC at diagnosis who convert to the RAS wildtype using monitoring of the RAS mutation status by liquid biopsy. Liquid biopsy may be also suitable for therapy response and for the detection of early signs of therapy resistance. Furthermore, the application of liquid biopsy may also help to reduce the long turnaround time of one month from biopsy of the tumor tissue to completion of the molecular profile. One of the main limitations of this study was the relatively long turnaround time between biopsy and completion of molecular analysis and between initiation of molecular profiling and discussion of the MDT and molecular-based therapy initiations with 30 and 42 days, respectively.

Time is a highly critical factor in the therapeutic management of mCRC, and a turnaround time of over one month without administration of effective therapy means that the mCRC progresses further. The growing metastases, particularly liver metastases, may lead to liver failure, increasing bilirubin values, and rapid health deterioration, making administration of the recommended targeted therapy impossible. From 28 molecularly targeted therapy recommendations, less than half of the patients eventually received the therapy. The TTF of 3.1 months and the disease control rate of 10% are modest outcomes. One reason for these modest outcomes may be that, due to the long turnaround time, there was not enough time for the targeted therapy to display its full potential. Other reasons may be the aforementioned tumor heterogeneity and the non-consideration of other aspects of the molecular profile as an important limitation. Interestingly, the three patients who achieved therapy response and one of the three patients who had stable disease experienced also improvements in their quality of life due to improvements in tumor pain intensity.

Furthermore, the design of this study is retrospective and it may be associated—in contrast to a prospective randomized controlled trial—with several limitations, including selection bias, insufficient documentation of clinical data, inadequate consideration of potential confounders, and the lack of randomization.

An additional limiting factor is that this retrospective study was conducted at a single center with a relatively small number of patients. It is difficult to demonstrate treatment efficacy, treatment difference, or certain findings in a small sample. In addition, single-center studies lack external validity to support or confirm the findings. Further research and clinical trials are warranted to evaluate the role and value of precision medicine for the management of mCRC patients.

In our study, we paid close attention to potential gender-specific differences. We did not find any gender-specific differences regarding the 28 targeted therapy recommendations.

Our study emphasizes the relevance and efficacy of pembrolizumab, even in metastatic patients with heavily pretreated therapy refractory mCRC who were classified as MSI-H.

It was shown that MSI-H status is a favorable prognostic factor at early local stages, particularly in stage II [31]. However, in stage IV CRC, MSI-H confers an inferior prognosis when compared to microsatellite stable metastatic CRC [32,33,34].

In our limited subset of patients with MSI-H, we administered pembrolizumab to four patients as no other druggable target could be derived from the molecular profile. Although MSI-H is not a favorable predictive marker in stage IV CRC, we achieved an impressive response with one complete remission, two partial remissions, and one stabile disease.

Pembrolizumab was the first tissue-agnostic treatment that was granted approval by the Food and Drug Administration (FDA) by mid-2017 for patients with unresectable or metastatic MSI-H or with mismatch repair deficient (dMMR) solid tumors that progress following prior treatment and who have no satisfactory alternative treatment options or those with MSI-H or dMMR colorectal cancer that progresses following treatment with a fluoropyrimidine, oxaliplatin, and irinotecan [35]. However, its use for MSI-H patients has not been still approved by the European Medicines Agency (EMA), and it was applied in off-label form in this study.

Taken together, the management of mCRC patients poses several major challenges, including the long turnaround time and the complex molecular heterogeneity of CRC. Our study underscores the relevance of immunohistochemistry and underlines the importance of time as a highly critical factor in precision medicine. Based on our study, molecular-based treatment approaches can be of clinical benefit in select heavily pretreated mCRC patients. In this study, the overall benefit for precision medicine approaches was limited and the TTF was relatively modest. However, precision medicine is a rapidly evolving field. In the next few years, technical advances will allow us to employ larger gene panels to cover and identify more mutations, amplifications, deletions, and gene fusions in a shorter period of time. The development of new and potent molecularly targeted therapies together with technical progresses in molecular profiling allow us to hope that, in the future, we may be able to yield deep and durable responses in heavily pretreated mCRC patients.

## Figures and Tables

**Figure 1 jpm-10-00272-f001:**
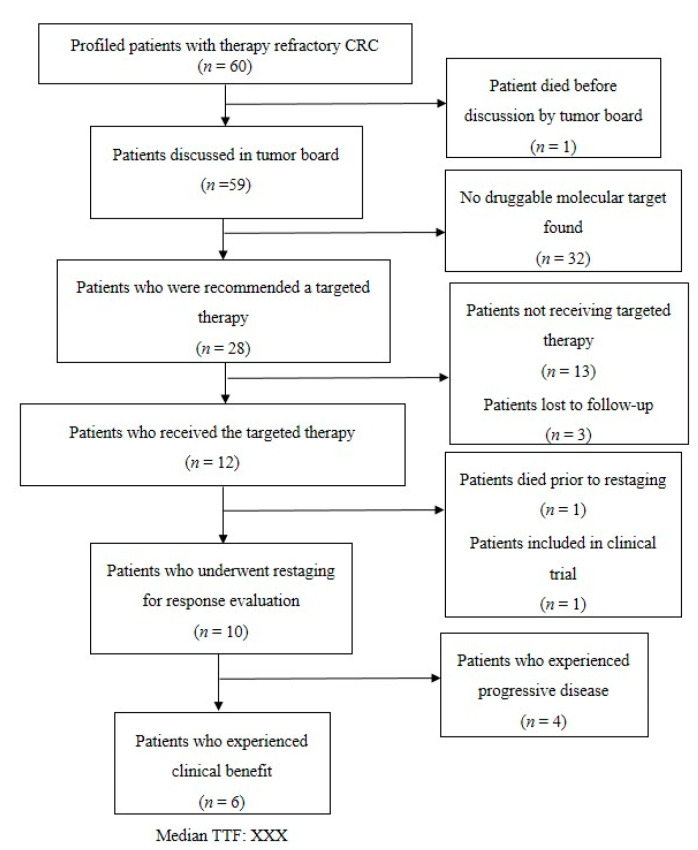
Flow chart of the 60 metastatic colorectal cancer (mCRC) patients.

**Figure 2 jpm-10-00272-f002:**
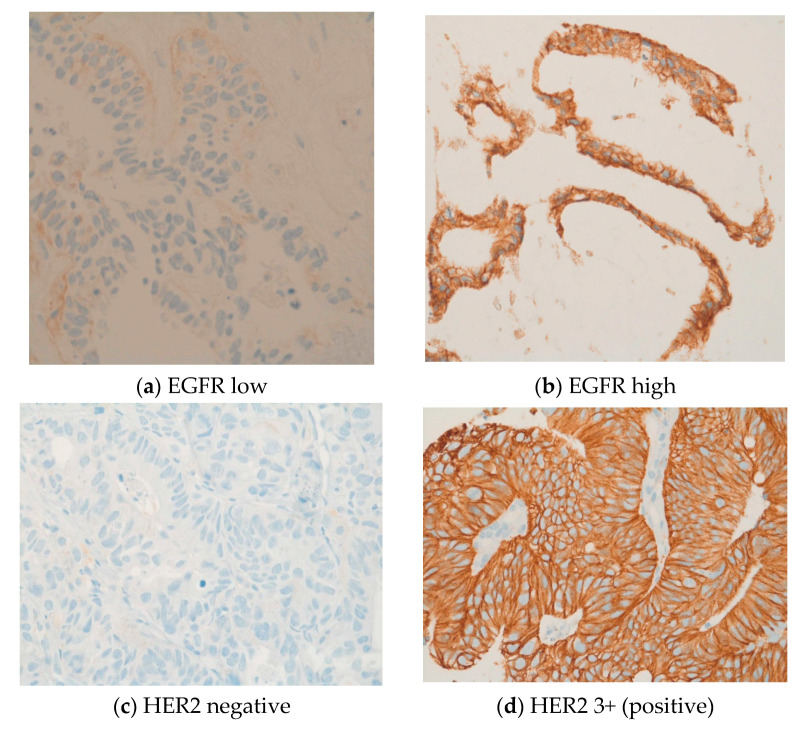
(**a**–**n**) These original images of immunohistochemistry show the differences between low and high expressions of various markers (Images by kind courtesy of Professor Dr. Müllauer).

**Figure 3 jpm-10-00272-f003:**
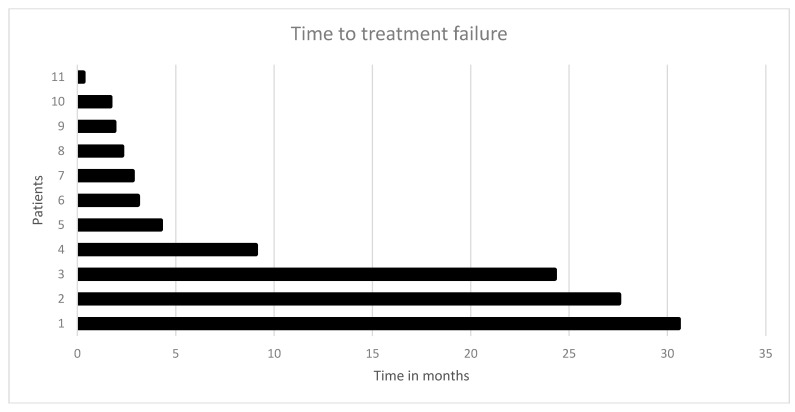
Time to treatment failure (TTF) in 11 CRC patients who received the recommended targeted therapy: the median time to treatment failure was 3.1 months.

**Figure 4 jpm-10-00272-f004:**
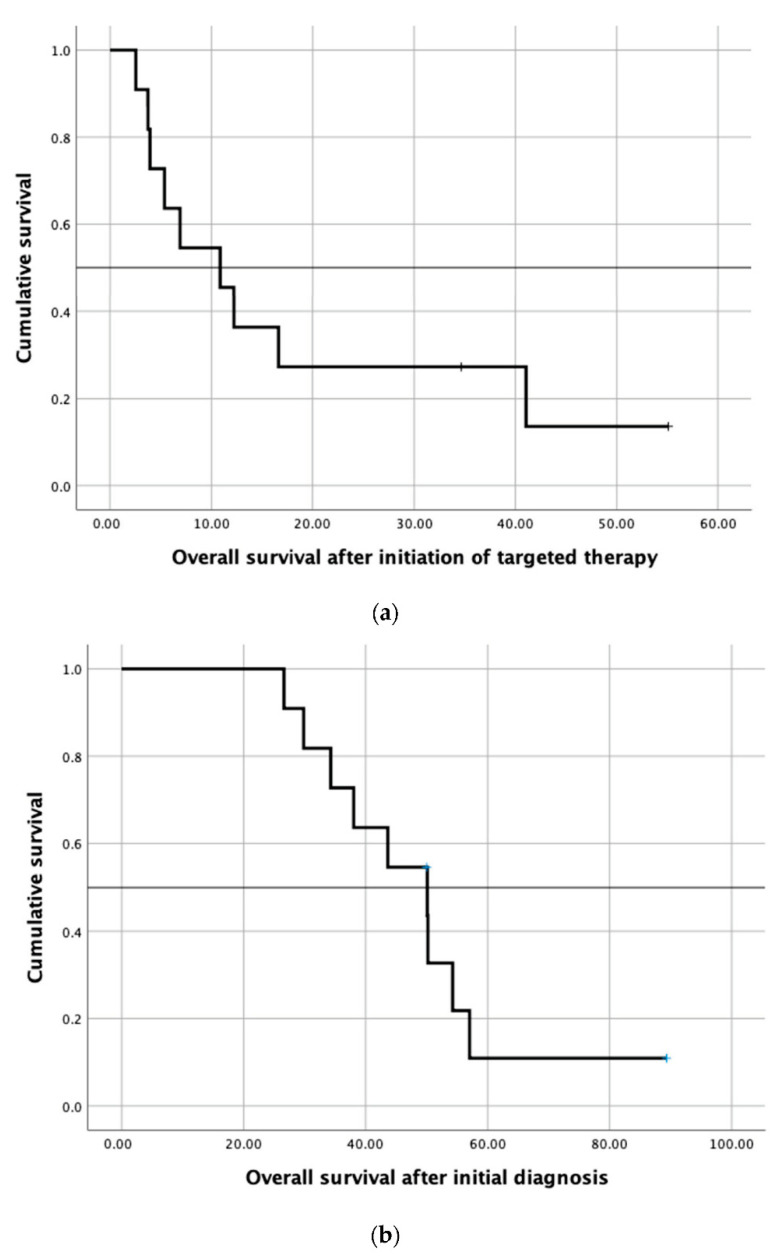
(**a**) Kaplan–Meier survival curve showing overall survival after initiation of targeted therapy in twelve patients receiving the targeted therapy and (**b**) Kaplan–Meier survival curve showing overall survival after initial diagnosis of mCRC in twelve patients receiving the targeted therapy.

**Table 1 jpm-10-00272-t001:** Patient characteristics (*n* = 60).

Patient Characteristics	Number
Median (range) age in years at first diagnosis	54.9 (16.9–81.2)
Median (range) age in years at time of molecular profiling	57.9 (19.3–84.3)
Male patients	42 (70%)
Female patients	18 (30%)
Caucasian	60 (100%)
Relapsed disease	23 (38%)
Metastatic disease	60 (100%)
Systemic chemotherapy received	60 (100%)
Prior chemotherapy regimens	2–6
Targeted therapy recommendations for patients for male patients for female patients	28 (47%) 19 (32%) 9 (15%)
Colorectal cancer localization Right ○ Cecum ○ Ascending colon ○ Transverse colon Left ○ Descending colon ○ Sigmoid colon ○ Rectum	13 (22%) 6 (10%) 5 (8%) 2 (3%) 47 (78%) 2 (3%) 27 (45%) 18 (30%)
Number of metastasis	106
Liver metastasis	46 (43%)
Lung metastasis	32 (30%)
Peritoneal carcinomatosis	17 (16%)
Bone metastasis	4 (4%)
Cerebral metastasis	3 (3%)
Cutaneous metastasis	2 (3%)
Renal metastasis	1 (1%)
Splenic metastasis	1 (1%)
Number of mutations detected	166

**Table 2 jpm-10-00272-t002:** Genomic profile of the therapy refractory CRC patients (*n* = 60).

Mutated Genes	Number of Mutations	Percentage of Occurrence in Patients (*n* = 60)	Percentage of all Mutations (166 Mutations in Total)
*TP53*	36	60.0%	21.7%
*KRAS*	29	48.3%	17.5%
*APC*	15	25.0%	9.0%
*PIK3CA*	9	15.0%	5.4%
*PTEN*	8	13.3%	4.8%
*ATM*	5	8.3%	3.0%
*SMAD4*	5	8.3%	3.0%
*NOTCH1*	4	6.7%	2.4%
*BRCA2*	3	5.0%	1.8%
*FBXW7*	3	5.0%	1.8%
*FGFR3*	3	5.0%	1.8%
*PTCH1*	3	5.0%	1.8%
*ERBB4*	2	3.3%	1.2%
*FANCA*	2	3.3%	1.2%
*GNAS*	2	3.3%	1.2%
*NOTCH3*	2	3.3%	1.2%
*POLE*	2	3.3%	1.2%
*SLX4*	2	3.3%	1.2%
*ALK*	1	1.7%	0.6%
*AR*	1	1.7%	0.6%
*ARID1A*	1	1.7%	0.6%
*ATRX*	1	1.7%	0.6%
*BRAF*	1	1.7%	0.6%
*CCND1*	1	1.7%	0.6%
*CDK12*	1	1.7%	0.6%
*CREBBP*	1	1.7%	0.6%
*CTNNB1*	1	1.7%	0.6%
*EGFR*	1	1.7%	0.6%
*ESR1*	1	1.7%	0.6%
*IDH1*	1	1.7%	0.6%
*JAK3*	1	1.7%	0.6%
*KDR*	1	1.7%	0.6%
*KIT*	1	1.7%	0.6%
*MAP2K1*	1	1.7%	0.6%
*MRE11A*	1	1.7%	0.6%
*MYC*	1	1.7%	0.6%
*NF1*	1	1.7%	0.6%
*NF2*	1	1.7%	0.6%
*NRAS*	1	1.7%	0.6%
*NTRK3*	1	1.7%	0.6%
*PALB2*	1	1.7%	0.6%
*RAD50*	1	1.7%	0.6%
*RB1*	1	1.7%	0.6%
*RICTOR*	1	1.7%	0.6%
*RNF43*	1	1.7%	0.6%
*SMARCA4*	1	1.7%	0.6%
*SMARCB1*	1	1.7%	0.6%
*SMO*	1	1.7%	0.6%
*TSC2*	1	1.7%	0.6%

**Table 3 jpm-10-00272-t003:** Rationale for targeted therapy recommendations.

Therapeutic Agent (Trading Name)	Targets	Overview of Current FDA Approval in Different Entities	Overview of Current EMA Approval in Different Entities	Number of Recommended and Received Cases and Responses
Pembrolizumab (Keytruda^®^)	PD-1, hypermutability	Melanoma, NSCLC, HNSCC, HL, Urothelial carcinoma, microsatellite instability-high cancer, gastric cancer, cervical cancer	Melanoma, NSCLC, HNSCC, HL, Urothelial carcinoma	-Recommended for and applied in 4 patients with MSI-H status: 1 patient achieved SD for 4.2 months; 2 patients achieved PR for 24.3 months and 30.5 months, respectively; 1 patient achieved CR for 27.5 months
Cetuximab (Erbitux^®^)	EGFR	CRC, HNSCC	CRC, HNSCC	-Recommended in combination with everolimus for 1 patient with EGFR expression and *KRAS* wildtype (patient initially had a KRAS mutation) and loss of PTEN and mTOR expression -Recommended in combination with irinotecan for 1 patient with EGFR expression and the *KRAS* wildtype (patient had initially a KRAS mutation)
Vemurafenib (Zelboraf^®^)	BRAF V600E	BRAF V600E melanoma or NSCLC BRAF V600E melanoma	BRAF V600E melanoma or NSCLC BRAF V600E melanoma	-Recommended for 1 patient with BRAF V600E (recommended prior to the clinical phase III BEACON trial)
Nintedanib (Vargatef^®^, Ofev^®^)	FGFR, FLT3, PDGFR, VEGFR	Idiopathic pulmonary fibrosis	NSCLC	-Recommended for 3 patients: 1 patient had a FLT3 amplification, 1 patient had a FGFR3 fusion gene, and 1 patient had PDGFRA expression
Vismodegib (Erdivedge^®^)	SMO	Basal cell carcinoma	Basal cell carcinoma	-Recommended for 1 patient with the SMO mutation
Everolimus (Afinitor^®^)	mTOR expression	Breast cancer, PNET, RCC, renal angiomyolipoma, subependymal giant cell astrocytomas (SEGAs) with tuberous sclerosis complex (TSC)	Breast cancer, RCC, Neuroendocrine tumors of pancreatic, gastrointestinal, or lung origin	-Recommended for 6 patients with strong p-mTOR expression and PTEN deficiency: in one case, it was recommended and applied in combination with raltitrexed. The patient achieved SD for 9.0 months. In two cases, it was recommended and applied in combination with bevacizumab. Both patients experienced PD. In one case, it was recommended in combination with Cetuximab.
Trastuzumab (Herceptin^®^)	HER2	HER2+ breast cancer and gastric cancer	HER2+ breast cancer and gastric cancer	-Recommended for and applied in combination with lapatinib for 2 HER2+ patients: 1 patient achieved SD for 1.9 months and 1 patient experienced PD.
Lapatinib (Tykerb^®^, Tyverb^®^)	HER2, EGFR	HER2+ breast cancer	HER2+ breast cancer	-Recommended for and applied in combination with trastuzumab for 2 HER2+ patients: See trastuzumab.
Afatinib (Gilotrif^®^)	*EGFR*, HER1, HER2, HER3	NSCLC	NSCLC	-Recommended for 5 patients with HER3 expression and applied in 1 patient. The patient experienced PD.
Crizotinib (Xalkori^®^)	ALK, ROS1, HGFR, MET	ROS1+ or ALK+ NSCLC	ROS1+ or ALK+ NSCLC	Recommended for 2 patients with MET expression
Erlotinib (Tarceva^®^)	EGFR	NSCLC, PDAC	NSCLC, PDAC	-Recommended for 1 patient with the EGFR mutation
Sunitinib (Sutent^®^)	PDGFR, KIT, VEGFR, RET, FLT3	RCC, PDAC, GIST	RCC, PDAC, GIST	-Recommended in combination with capecitabine for 1 patient with the KIT mutation: the patient was enrolled in the phase II SUNCAP trial.

ABL1, Abelson murine leukemia viral oncogene homolog 1; AML, acute myeloid leukemia; ALL, acute lymphatic leukemia; BCR, breakpoint cluster region; CML, chronic myeloid leukemia; CRC, colorectal cancer; EGFR epidermal growth factor receptor; EMA, European Medicines Agency; FDA, Food and Drug Administration; FLT3, fms like tyrosine kinase 3; GIST, gastrointestinal stromal tumor; GNRHR, gonadotropin-releasing hormone receptor; HER2, human epidermal growth factor receptor 2; HL, Hodgkin lymphoma; HNSCC, head and neck squamous cell carcinoma; MCL, mantle cell lymphoma; MDS, myelodysplastic syndrome; MPD, myeloproliferative disorder; NSCLC, non-small-cell lung carcinoma; PD, progressive disease; PD-1, programmed cell death protein 1; PDAC, pancreatic ductal adenocarcinoma; PDGFR, platelet-derived growth factor receptor; Ph+: Philadelphia chromosome positive; p-mTOR, phosphorylated mammalian target of rapamycin; RCC, renal cell carcinoma; RET, rearranged during transfection; SD, stable disease; VEGFR, vascular endothelial growth factor.

**Table 4 jpm-10-00272-t004:** Characteristics of the CRC patients receiving the molecular-based targeted therapy recommendation (*n* = 12).

Number, Gender, Localization of CRC	Detected Mutations; Gene Fusions	Immunohistochemistry	Applied Targeted Therapy	Age (in Years) at Time of Molecular Profiling	TTF in Months	Therapy Response	Cause of Therapy Termination
1 Male Sigmoid carcinoma	*KRAS*	EGFR score = 90, MET score = 3, p-mTOR score = 110, loss of PTEN	Everolimus combined with bevacizumab	65.5	2.8	PD	PD
2 Male Sigmoid carcinoma	*NRAS, PTEN*	p-mTOR score = 65, loss of PTEN	Everolimus combined with raltitrexed	49.0	9.1	SD	PD
3 Male Sigmoid carcinoma	No mutations detected	MSI-H	Pembrolizumab	53.2	24.3	PR	PD
4 Male Carcinoma of the ascending colon	*APC, PTEN, TP53*	MSI-H, EGFR score = 180, MET score = 3, p-mTOR score = 80	Pembrolizumab	48.7	30.6	PR	PD
5 Male Cecum carcinoma	*APC, KRAS, TP53*	EGFR score = 30, HER3 score = 3, MET score = 2, p-mTOR = 100,	Afatinib	55.6	2.3	PD	PD
6 Male Rectal cancer	*PIK3CA, TP53*	EGFR score = 280, HER2 score = 2, HER3 score = 3, MET score = 2, p-mTOR = 240	Trastuzumab combined with lapatinib	56.2	1.9	SD	PD
7 Male Sigmoid carcinoma	*AR, ARID1A, ATM, PALB2, PIK3CA, RNF43;* *FGFR3—TACC3*	MSI-H EGFR score = 200, PTEN score = 100, p-mTOR = 240,	Pembrolizumab	58.3	4.2	SD	PD
8 Female Sigmoid carcinoma	*CTNNB1, KRAS, PTEN, RB1, TP53,*	MSI-H,	Pembrolizumab	42.0	27.5	CR	Relapse
9 Female Sigmoid carcinoma	No mutations detected	HER2 score = 3, p-mTOR = 120	Trastuzumab combined with lapatinib	50.3	3.1	PD	PD
10 Female Rectal cancer	*BRCA2, KRAS, POLE, PTCH1, RAD50, TP53*	EGFR score = 50, p-mTOR = 60,	Everolimus combined with bevacizumab	57.0	1.7	PD	PD
11 Female Sigmoid carcinoma	*TP53*	EGFR score = 110, HER3 score = 2, MET score = 1, p-mTOR = 240, loss of PTEN	Everolimus	19.3	0.3	n.a.	Death
12 Male Carcinoma of the ascending colon	*KIT, KRAS, TP53*	EGFR score = 50, MET score = 2, p-mTOR = 40,	Sunitinib combined with capecitabine; The patient was enrolled in the phase II SUNCAP trial.	70.0	n.a.	n.a.	n.a.

n.a., not applicable; AR, androgen receptor; CPS, combined prognostic score; ECOG PS, Eastern Cooperative Oncology Group performance status; EGFR epidermal growth factor receptor; MSI-H, microsatellite instability-high; PD, progressive disease; PDL1, programmed death ligand 1; PDGFRA, platelet-derived growth factor receptor alpha; p-mTOR, phosphorylated mammalian target of rapamycin; SD, stable disease, PTEN, phosphatase and tensin homolog; TPS, tumor positive score.

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
