# Peer review of "Precision Medicine for the Management of Therapy Refractory Colorectal Cancer"

_jpm, 2020, doi:10.3390/jpm10040272_

Round 1
Reviewer 1 Report
I read with interest the manuscript by Taghizadeh et al. This retrospective single center study, describes the implementation and local results of a precision medicine platform for metastatic colorectal cancer. The article is well written and is interesting. I only have some minor comments:
- There is not description of what outcomes were measured and how were they measured. The authors report "stable disease", "partial response" and "treatment failure" in the results section, but these terms need to be clearly explained in the methodology section.
- After the molecular-based therapy was started, did patient undergo a standardized follow-up? Can you describe more in the methodology section how were patients follow-up, and how was death ascertained? Instead of reporting median time to treatment failure, the authors should consider presenting survival analysis with a Kaplan Meir curve, which is more standardized for this type of analysis
- The first paragraph of the methodology should clearly stipulate that this was a single center retrospective study, where was it conducted, and the years of the study period.
- The results are packed into a very large paragraph. The authors should consider splitting up the results into different sections.
- The authors mention the median time between the molecular profile and the multidisciplinary team discussion for those who ultimately received MTB (n=12). How about for the patients who did not receive the drug? If the discussion was delayed in this group, this may have impacted outcomes and it's worth of mentioning it.
- The median time to failure presented in the results says 0.3-30.6 months. Is this a range or interquartile range? This should be clarified
- Table 1 should mention the location of CRC metastasis
- The discussion does not mention about any of the limitations of the study. A paragraph with study limitations need to be included. Problems with retrospective design and single center studies. Problems to interpret outcomes in such a small sample (n=12), which makes it more of a case series in that regards. Rapid change of targets and drugs in this area.
Author Response
Thank you very much for your valuable comments and important suggestions!
- In the methodology section of the revised version, we defined the terms complete response, partial response, stable disease and progressive disease and treatment failure. See lines 114-127 on pages 5 and 6.
- Follow-up was done every 8 to 12 weeks for outcome evaluation by radiological assessment - depending on the respective therapy. If the patient did not appear on the follow-up date, we searched our electronic data processing system that is linked to the national death register to check and ascertain the death of the patient in the meantime. See lines 128-131 on page 6.
Further, we provide in the revised version of our article Kaplan Meier survival curves. See page 20. The median time to treatment failure (TTF) in patients who received the targeted therapy was 3.1 months (range: 0.3–30.6 months; see Figure 2 and Table 4). The median overall survival (mOS) of these 12 patients after initial diagnosis of mCRC was 50.1 months. The mOS after initiation of targeted therapy was 10.9 months (see Figures 3a and 3b). See lines 313-316 on page 12. - The first paragraph of the revised methodology section now clearly states that this was a retrospective single center study and we also mentioned – the location and study period of this study. See lines 96-101 on page 5.
- In the revised article, the results have been splitted into different sections to improve readability.
- The median time interval between the initiation of molecular profiling and discussion of the MDT and molecular-based therapy initiation for the other patients (n= 48) was 29 days. Thus, it was even slightly shorter than the twelve patients receiving the targeted therapy. See lines 262-264 on page 10.
- We clarified the median time to treatment failure: It is a range. See line 318 on page 12.
- In the revised version, we mentioned location of CRC metastasis. See Table 1 on page 13.
- The revision discussion now clearly points out the limitations of this study. See pages 21-23.
Reviewer 2 Report
The authors have attempted to combine clinical, pathological, genomics and IHC data to identify candidates that can be targeted to provide therapeutic benefit in mCRCs. This is a good attempt but needs to be refined further both in terms of the results displayed, order of the results and overall discussion of their findings.
Results:
Line 229 – The median age…. The number in the text does not represent the number in table 1. Authors to check and fix.
It is unclear if the biopsy sample was before any of treatment administration or post treatment? This needs to be stated explicitly.
What were the lines of prior palliative systemic chemotherapies that the patients received. This information needs to be provided.
The authors report that seven gene – fusions were identified, however, do not mention using which type of data? Was it from the mutation panel?
Gene amplification? How did they find these, what method was used?
Examples of IHC (showing positive, negative controls) should be provided for the staining that they report e.g mTOR, EGFR, Her2+, PTEN. Also as they are scoring these slides, they should give examples of images representative of the low and high based on their scoring criteria.
For PTEN where they have used FISH, example of this needs to be provided.
Lines 257 – 259…What is common expression? is it high, low or expected?
Flowchart should be used first and will be very useful in understanding the approach taken by the MTB
Why were only 12 patients recommended targeted therapies? Did the other patients not qualify? A reason needs to be provided?
Where in Table 3 is the rationale provided? Table 3 is more like a literature report of what is known about these drugs and does not add much value to the manuscript, could be moved to supplementary information.
Table 4 needs to be cited in the text. Also it is very informative.
The authors also stained for PD-L1, CD20 etc, why have they not provided any information regarding these markers? If they are not planning on disclosing this information, then they need to modify the methods to remove these markers.
Discussion:
“prime example of this phenomenon is the recommendation of cetuximab for two patients who were initially KRAS-mutated and were therefore not treated with an anti-EGFR therapy; however, at the time of molecular profiling, they had developed the RAS wildtype” .. How does the authors, only looking at the intial biopsy material justify the treatment that will benefit the metastasized tumours? Would it not be better to molecularly profile the metastatic CRCs as well? The authors should provide some discussion around the this.. maybe lack of benefit can be partly attributed to a different molecular landscape of the metastatic versus primary tumour.
Is 3.1 months enough to go through extra targeted treatment? What about the quality of life versus impact of treatment? This needs to be discussed.
What about the role of the immune system? Also it is well established that MSI tumours do have good outcomes, what about non-MSI tumours?
Are 60 patients of which only 12 went onto get targeted treatment statistically enough to make any conclusions?
Overall the discussion needs to give more details….
Author Response
Thank you very much for your ideas and thank you for the gracious observations!
- Thank you very much for pointing that out! We corrected the mistake. See line 253 on page 10.
- Biopsy was performed after failure of all standard treatment options. See line 256 on page 10.
- In the revised version, we mentioned the palliative therapy regimens. See lines 270-273 on page 10.
- The 161-gene next-generation sequencing panel of Oncomine Comprehensive Assay v3 (Thermo Fisher Scientific, Waltham, MA, USA), which covers genetic alterations, gene amplifications and gene fusions. See line 146 on page 6.
- The 161-gene next-generation sequencing panel of Oncomine Comprehensive Assay v3 (Thermo Fisher Scientific, Waltham, MA, USA), which covers genetic alterations, gene amplifications and gene fusions. See line 146 on page 6.
- Unfortunately, the molecular pathology could not provide us examples of images of immunohistochemistry. However, all slides and specimens were examined and checked by an experienced and competent molecular pathologist who is also one of the co-authors of this study.
- We clarified the expression intensity. See line 292 on page 11.
- As requested, we put the flow chart first. See page 13.
- In the revised version, we explained the reason: The other 32 patients (53%) did not qualify for a targeted therapy due to the lack of actionable molecular targets. See lines 299-300 on 11.
- Table 3 explains the targets and also describes in the fifth column the molecular aberrations that were the basis for the targeted therapy and reports the responses. We would be very much obliged if we could leave Table 3 in the main text, as we believe that it increases clarity of the results.
- Table 4 is now cited two times in the manuscript: see line 309 on page 11 and line 319 on page 12.
- In the revised results, we also included the markers PD-L1 and KIT. The expression of other markers was not observed. See lines 293-295 on page 11.
- This is an important question: Tumor tissue used for molecular profiling was obtained by biopsy in 25 patients (42%) or during surgical treatment in the other 35 patients (58%). Biopsy was performed after failure of all standard treatment options. Whenever possible, we used metastatic tissue for molecular profiling, which was particularly suited when fresh biopsies were obtained. If this approach was not feasible, e.g. if the anatomic site was not suitable for biopsy, we used the information obtained from the primary tumor site. Despite potential spatial heterogeneity, we assumed that most of the genetic aberrations in the primary cancer were also present at the metastatic site. Studies have shown that there is a high biomarker concordance between primary colorectal cancer and its metastases. See page 21.
- The three patients who achieved therapy response and one of the three patients who had stable disease, experienced also an improvement in their quality of life due to an improvement of tumor pain intensity. See lines 312-314 on page 11 and page 22/23 in discussion.
- It was shown that MSI-H status is a favorable prognostic factor at early local stages, particularly in stage II. However, in stage IV CRC, MSI-H confers an inferior prognosis when compared to microsatellite stable metastatic CRC. See page 23.
- An important limitation of this study is that it was conducted at a single center with a relatively small number of patients. It is difficult to demonstrate treatment efficacy or treatment difference or certain findings in a small sample. In addition, single center studies lack external validity to support or confirm the findings. Further research and clinical trials are warranted to evaluate the role and value of precision medicine for the management of mCRC patients. See page 23.
- The revised discussion contains more details and limitations.
Round 2
Reviewer 2 Report
The revised version is much better, however, there are still concerns that need to be addressed:
What are the standard treatment options? It can differ between countries. Needs to be described in the Methods section.
The authors are not able to provide any examples which makes it impossible to assess how they have scored these samples. Only having a molecular pathologist as one of the co-authors is not sufficient to understand how the data was generated. Unfortunately this information has to be provided, else the results are not very meaningful. The authors also stained for PD-L1, CD20 etc, why have they not provided any information regarding these markers? If they are not planning on disclosing this information, then they need to modify the methods to remove these markers.
“It was shown that MSI-H status is a favorable prognostic factor at early local stages, particularly in stage II. However, in stage IV CRC, MSI-H confers an inferior prognosis when compared to microsatellite stable metastatic CRC. See page 23.” Yes this is true in the literature, how is this applicable to their own dataset? Have they looked at this?
“Studies have shown that there is a high biomarker concordance between primary colorectal cancer and its metastases.” They only cite one article, which is looking at specific markers alone. However, this paper does not provide evidence for the markers that the authors are looking at. Also where the authors have used the biopsies from mCRC or blood, have they done a comparison to see if the patients who benefited were those whose biopsies came from mCRC/blood compared to original tumour?
Author Response
1) What are the standard treatment options? It can differ between countries. Needs to be described in the Methods section.
Instead of the section methods, this information has been provided by the authors in the section results, where we felt that it better supported the flow of the manuscript (please refer to lines 270 to 274 on page 10 for a description of this issue).
2) The authors are not able to provide any examples which makes it impossible to assess how they have scored these samples. Only having a molecular pathologist as one of the co-authors is not sufficient to understand how the data was generated. Unfortunately, this information has to be provided, else the results are not very meaningful.
We tried our best and despite some technical problems we could provide – as requested – images of immunohistochemistry in the revised manuscript. Please refer to the Figures 3a – n on pages 14-15. Unfortunately, we could not obtain by the molecular pathology due to technical problems.
3) The authors also stained for PD-L1, CD20 etc., why have they not provided any information regarding these markers? If they are not planning on disclosing this information, then they need to modify the methods to remove these markers.
We provided this information in the first revision of our manuscript. Please refer to lines 284 to 295 on page 11 for a detailed description of this topic.
4) “It was shown that MSI-H status is a favorable prognostic factor at early local stages, particularly in stage II. However, in stage IV CRC, MSI-H confers an inferior prognosis when compared to microsatellite stable metastatic CRC. See page 23.” Yes, this is true in the literature, how is this applicable to their own dataset? Have they looked at this?
In the revised version we wrote: "In our limited subset of patients with MSI-H, we administered pembrolizumab to four patients as no other druggable target could be derived from the molecular profile. Although MSI-H is not a favourable predictive marker in stage IV CRC, we achieved an impressive response with one complete remission, two partial remissions and one stabile disease." Please refer to page 25.
5) “Studies have shown that there is a high biomarker concordance between primary colorectal cancer and its metastases.” They only cite one article, which is looking at specific markers alone. However, this paper does not provide evidence for the markers that the authors are looking at. Also, where the authors have used the biopsies from mCRC or blood, have they done a comparison to see if the patients who benefited were those whose biopsies came from mCRC/blood compared to original tumour?
We wrote in the revised results section: “Three tumor specimens from the ten patients who underwent radiological assessment were obtained during a conventional tumor biopsy. Two of these three patients experienced progressive disease, one patient achieved stable disease." Please refer to lines 326 to 328 on page 12.
We wrote in the revised discussion: “However, the number of patients (n= 10) who underwent radiological response assessment after application of targeted therapy was too limited to examine the influence of the biomarker concordance on the outcome. Thus, further clinical trials and studies are required to examine the degree of biomarker concordance between primary cancer and metastases.” Please refer to pages 23-24.
Round 3
Reviewer 2 Report
Addition of the data and modification to the text has substantially improved the manuscript. No further issues need to be addressed.
Author Response
Dear Reviewer,
Thank you very much for your comments and ideas.
Yours sincerely from Vienna
Hossein Taghizadeh